# Complex Physical Properties of an Adaptive, Self-Organizing Biological System

**József Prechl** 

R&D Laboratory, Diagnosticum Zrt., H-1047 Budapest, Hungary; jprechl@diagnosticum.hu

**Abstract:** Physical modeling of the functioning of the adaptive immune system, which has been thoroughly characterized on genetic and molecular levels, provides a unique opportunity to define an adaptive, self-organizing biological system in its entirety. This paper describes a configuration space model of immune function, where directed chemical potentials of the system constitute a space of interactions. A mathematical approach is used to define the system that couples the variance of Gaussian distributed interaction energies in its interaction space to the exponentially distributed chemical potentials of its effector molecules to maintain its steady state. The model is validated by identifying the thermodynamic and network variables analogous to the mathematical parameters and by applying the model to the humoral immune system. Overall, this statistical thermodynamics model of adaptive immunity describes how adaptive biological self-organization arises from the maintenance of a scale-free, directed molecular interaction network with fractal topology.

**Keywords:** system; network; chemical potential; thermodynamic activity; antibody; self-organization; fractal

## 1. Introduction

Macromolecules, and, in particular, proteins, have evolved as part and basis of the evolution of life. The diversity of proteins reflects the diversity of life itself, while the tree of evolution of proteins reflects the evolution of life [1]. Proteins serve to build (structural proteins) and operate (enzymes, transporters, regulators, secretions) an organism. For all these functions, it is necessary for proteins to interact with other molecules, from metal ions through small molecules to other macromolecules. Beyond spatial arrangement, that is, the necessity of being at the same place at the same time, interactions require thermodynamic probability for the binding. In other words, the system searches for a free-energy-minimum state while proteins sample interacting partners. This, in turn, means that most proteins have energy minimums in a bound state. The evolution of proteins from this aspect is the search for binding partners; genetic changes resulting in new, modified proteins are sustained if the binding is advantageous for the survival of the organism. Since paired interactions create networks of interactions, evolutionary processes are constrained by energy transduction networks [2]. These biological networks may mediate metabolic fluxes [3], protein interactions [4] or information in neural networks [5].

The functioning of the adaptive immune system is based on the directed evolution of proteins, called antigen (Ag) receptors, of lymphocytes [6,7]. In its essence, the adaptive immune system is a catabolic system; it removes molecules and cells from the body based on molecular, immunological instructions. Humoral adaptive immunity utilizes glycoproteins called antibodies (Abs) for tagging targets for removal [8]. Abs, unlike all other somatic proteins, are not encoded in the genome but are produced as a result of genetic recombination and mutation events during our lifetime (sometimes referred to as accelerated evolution). The role of the immune system is to drive and direct the evolution of these molecules so as to maintain the molecular and cellular integrity of the host; this is what we call immunity. As a result, millions of distinct Abs are produced constantly in our

bodies, removing cellular debris, maintaining balance with the microbiota and fending off invading agents [9].

On the molecular level, these immunological events have been characterized in detail; however, the systems-level understanding and physical description of the self-organization of the adaptive immune system are still incomplete. Since the adaptive immune system is a *bona fide*, complex, adaptive system, considering its number of elements, its diversity and its self-organizing capability, this bigger picture can be approached by characterizing immunological phenomena using the terminology of the physics of complex systems and mathematical models. Following the pioneering work of Perelson [10–12], the application of physical and mathematical models for the answering of questions about immune repertoire size and diversity and lymphocyte population dynamics, immunological memory has regained interest [13–17]. Here, I attempt to combine chemical thermodynamics with network theory, building up a model that is consistent with a novel technological approach to serum Ab reactivity measurement [18]. The basic concept of this model was introduced in previous papers that showed how stages of B-cell differentiation correspond to the generation of a thermodynamic system and a network of interactions [19,20]. In this paper, we first define the biological system as a configuration space using physical chemistry in order to identify analogues of cells and molecules in a physical system. Then, we use the mathematical description of the system to derive the architecture of the hierarchical network that governs Ag transport in a stationary state of the system. Finally, we discuss the model from the perspective of immunology, physical chemistry, network science and the physics of complex systems. The mathematical approach described connects these fields of science and, therefore, contributes to the quantitative, biological understanding of an adaptive biological system.

## 2. Materials and Methods

A mathematical approach was used to examine exponential, Gaussian and power law functions that serve as the basis of the proposed theoretical model; transformations of these functions revealed the relationships between thermodynamic, network science and immunological concepts building on such functions. The revealed analogies were used for validating the model.

## 3. Results and Discussion

The following sections first outline the theoretical aspects of the model and describe the mathematical approach taken. Then, support for the validity of the mathematical approach is provided by identifying the thermodynamic, network and immunological phenomena explained by the model.

### 3.1. Compartments of the System

Throughout the article, we examine the adaptive immune system in an abstract space called configuration space. Configuration here refers to the arrangement of all the constituent Abs in a system of interacting molecules. A general Ab structure is modified by the arrangement of atoms that contribute to the binding site of the Abs, creating a huge, conformational landscape. Since the formation of non-covalent bonds between interaction partners requires shape congruency, arrangement in this space determines both Ab and Ag structure and binding specificity and strength. The manifold enclosing components of the immune system in this configuration space thus define both Abs and Ags. In this space, cells and molecules of the immune system, corresponding to clones bearing or being a particular Ab, respectively, are positioned according to their potential to interact with a target and in the direction of the target. The interaction potential is the chemical potential of the Ab in body fluids. Targets are Ag molecules that drive the evolution of the system. Considering a three-dimensional Euclidean space as the configuration space of the system, target Ag shapes—epitopes—form a continuity on an imaginary, spherical canvas enclosing the system (Figure 1).

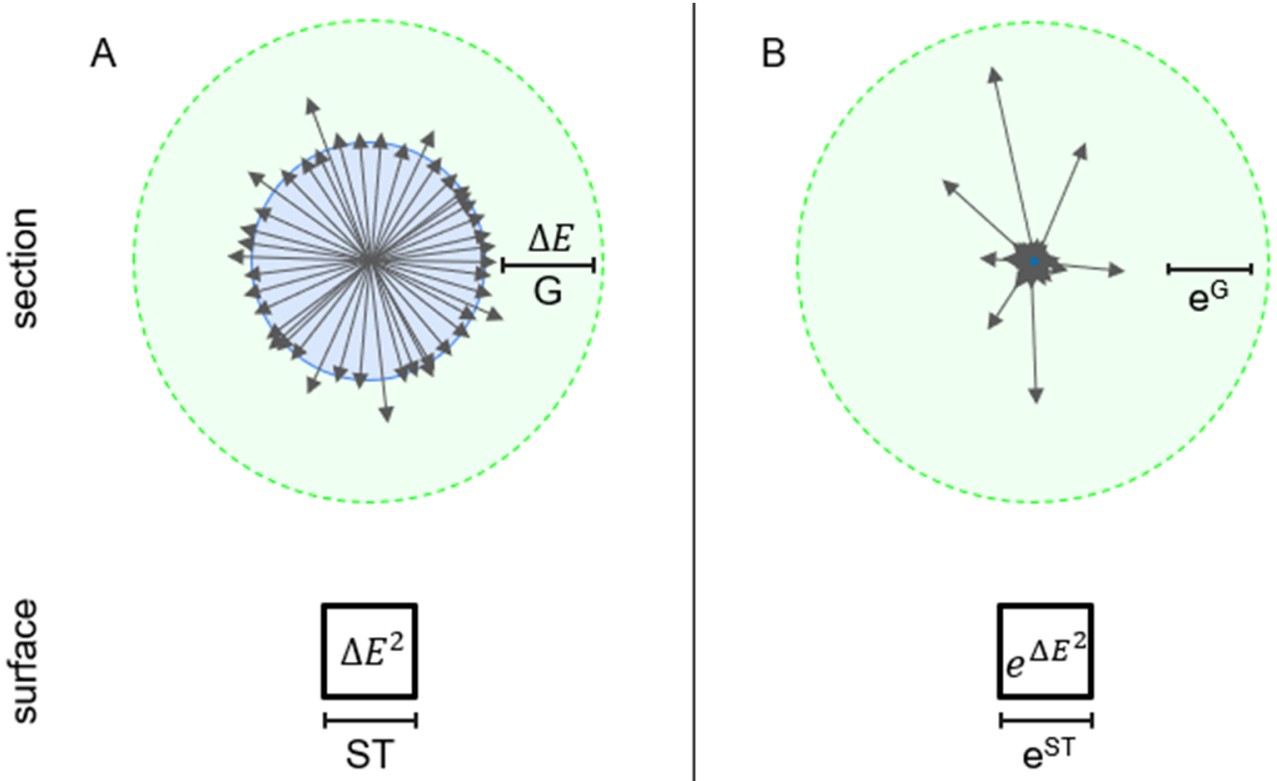

**Figure 1.** Compartments of the self-organizing system shown schematically in two scales: (**A**), energy scale, (**B**), thermodynamic activity scale. System core is blue, arrows represent chemical potentials of system components in interaction space, shown in green. Arrows represent vectors with the length as chemical potential and direction as specificity. G, Gibbs free energy; S, entropy; T, temperature; E, energy.

From a physical point of view, the system is embedded in a reservoir, which serves as a heat and particle bath; this is the host organism that maintains constant temperature and pressure of the system. The system is in a stationary state in the sense that it is constantly dispersing energy while maintaining its state functions. Overall, we can regard the system as an ensemble of overlapping and hierarchically arranged thermodynamic ensembles.

We consider three compartments in our configuration space model: 1. the core of the system, surrounded by the core surface; 2. interaction space around the core, extending from the core surface to the system surface; and 3. the system environment (surroundings) beyond the system surface. We use two scales for examining and visualizing the system: the energy scale (Figure 1A), which is the logarithmic form of the interaction probability, or the thermodynamic activity scale (Figure 1B). The system core contains the conceptual energy source of the system that is located in the origin of the configuration space. Events within the core, therefore, can be thermodynamically unfavorable, consuming energy to maintain a continuous supply of particles in the interaction space. The core surface is spherical and is characterized by system components with average molecular free energy $\mu_0$.

The system is composed of Ab and Ag molecules. B cells, which carry sequence information, at various stages of differentiation, determine Ab structure and define clonotypes. Soluble Abs released by B cells mediate energy transfer by binding and releasing Ags. Energy corresponds to Ag molecules, which are transferred from lower- to higher-affinity Abs, with a corresponding decrease in the system free energy. These constituents are present both in the core and in the interaction space. Interaction space is defined and confined by the totality of chemical potential vectors, each of which represents the interaction energy

of a particular Ab clonotype. Radial geometrical distance in this configuration space is measured in units of chemical energy per molecule or chemical potential, while vector directions identify Ag structures and molecular shapes in the surroundings. From a network point of view, in the humoral adaptive immune system, nodes are basically Ab molecules, while links represent Ab structural similarities and, thereby, represent potential pathways of Ag transport.

### 3.2. Principle of Self-Organization

Self-organization is the ability of a system to arrange its constituents so as to adjust to its environment and also maintain a responsive state. Self-organizing systems tend to reach an optimal state associated with minimal interaction and dissipation [21–23]. The immune system adjusts the quality (intensive physical property) and number of molecules (extensive physical property) in the host by the coupling of a sensor and an effector mechanism [20]. The sensor mechanism is a B cell that survives only in the presence of signals triggered by the target molecule via an Ab displayed on the cell surface. The effector mechanism is the generation of molecules, soluble Abs, that bind both to the target molecule and to cells that remove the resulting complexes. Coupling means the balanced adjustment of the chemical potential $\mu$ of the Abs on the sensor surface and of the effector molecules against that molecule species. Biologically, it means that the sensor and effector B cells are genetically closely clonally related, expressing identical or similar Abs either on their surface or as secreted components. Relations in terms of network connectivity are determined not only by similarity but also by the similarity of interaction partners [24]. In the case of the humoral immune system, this means that Abs that are clonally not or only distantly related may contribute to the binding and elimination of a given Ag molecule. In other words, in a subnetwork of the system dedicated for a given Ag, distinct Ab clonotypes may be linked together in the network of Ag transfer pathways. The concept of clone sizes and corresponding Ab chemical potential and Ag thermodynamic activity, with corresponding chemical potentials, can be visualized by activity maps and energy diagrams (Figure 2). The thermodynamic activity of a particular Ab clone is related to the frequency of the cells belonging to that clone (clone size) and the serum concentration of Abs. The chemical potential of an Ab is the logarithm of its activity. Similarity is represented by the topology of the activity map, with closely related clones being neighbors. The thermodynamic activity of an Ag is related to its ability to stimulate the immune system and the corresponding chemical potential is its logarithm. Again, structural similarity is represented by location in the activity map.

Effector cells (plasma cells) secrete Abs, which bind Ags with an efficiency determined by chemical potential and Ab concentration. During the active phase of the immune response, the chemical potential of the effector molecule is raised, and more bound Ag is removed from the system. At the same time, the coupled increase in the sensor sensitivity allows the cell to survive in the presence of less Ag, and a steady state with higher chemical potential is established. The active phase of the immune response is followed by contraction, wherein memory cells, the long-lived sensors and effectors of the system, are selected and maintained. We assume that, in the resting phase of the immune response, a steady state is maintained regarding Ab and Ag concentrations. Since several different Abs with distinct chemical potentials may contribute to Ag free-energy adjustment, the system selects clones with the chemical potential and concentration required. Our argument here is that this selection process is governed by thermodynamic rules.

The cells of the adaptive immune system, a tissue that penetrates the whole organism, possesses a unique property that no other cells have; lymphocytes are capable of directing the evolution of proteins within the organism [15]. These proteins, called Ag receptors, are generated by genetic recombination and mutation and selection events, resulting in a diversity that well exceeds the diversity of all other proteins in the organism. Depending on the size of the host, lymphocyte numbers are in the range of $10^8$–$10^{12}$ cells [25,26]. Generation of diversity is a random molecular process, while the function of the adaptive

immune system is to adjust, by cycles of division and checkpoints of selection, lymphocyte specificity and affinity. These two features correspond to the direction and length of the vector in the configuration space. A steady state, which is immunological rest in our system, is reached by the adjustment of an intensive and an extensive property. The intensive property is the chemical potential of Abs, determined by a cell's genes and the Ab sequence and structure. The extensive property is diversity and numerosity of Abs, the coverage of the conformational space of the system surface (Figure 1). Thus, in a broader sense, balancing is between molar free energy and system entropy.

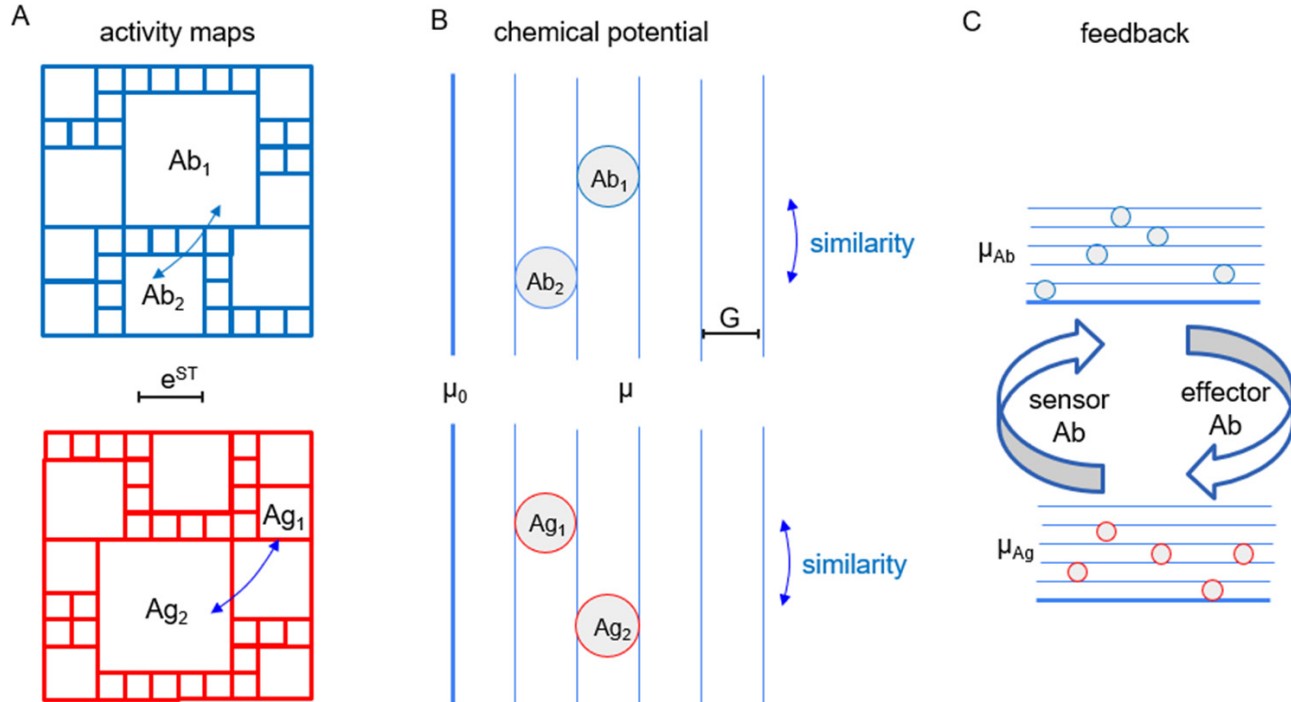

**Figure 2.** Activity and energy diagrams of antibodies and antigens. Thermodynamic activity, chemical potential and structural similarity appear in activity maps (**A**) and energy diagrams (**B**) for antibody (blue) and antigen (red). Square areas in the map represent activity; neighborhood distance (blue double arrows) means similarity. (**C**) Higher chemical potential results in better Ag-binding efficiency for effector Abs and higher Ag sensitivity for sensor Abs. Removal of Ags lowers signaling in sensor B cells; sustained Ag presence and immune stimulation triggers sensor B cells and triggers generation of effector Abs.

Self-organization also maintains responsiveness in the system. In our case, it is maintained by the sensors, B cells displaying membrane-bound Abs, that are able to initiate an immune response and, thereby, reset the chemical potential of both sensors and effectors and lead to the reorganization of local hierarchy in the system. Activity maps and energy diagrams in this paper, therefore, represent a particular point in the lifetime of a highly dynamic system.

### 3.3. Architecture of the System

The immune system is theoretically capable of interacting with any molecular structure. This means it takes up volume in all directions of Ag space, creating a spherical core (Figure 3). It has to allocate its limited resources for the generation of particles in the interaction space in the most energy-efficient way. Therefore, it is split up into a number of thermodynamic ensembles, distributing its resources between these. A particular ensemble is directed against a particular target epitope, adjusting its properties to attain a steady state. Because resources of the system are finite, ensembles compete for them and may overlap,

cluster and form hierarchies. At any one moment in time, the distribution of the state of the ensembles is the distribution of chemical potentials and their way of sharing system energy. The configuration space of the self-organizing system is a systemic ensemble in the sense that it is a collection of coexisting grand canonical ensembles, each ensemble responding to a different antigenic component of the environment. In the vector space of the chemical potentials of the system, the location of each ensemble identifies a particular direction, representing the potential energy of the interaction with the particular part of the environment.

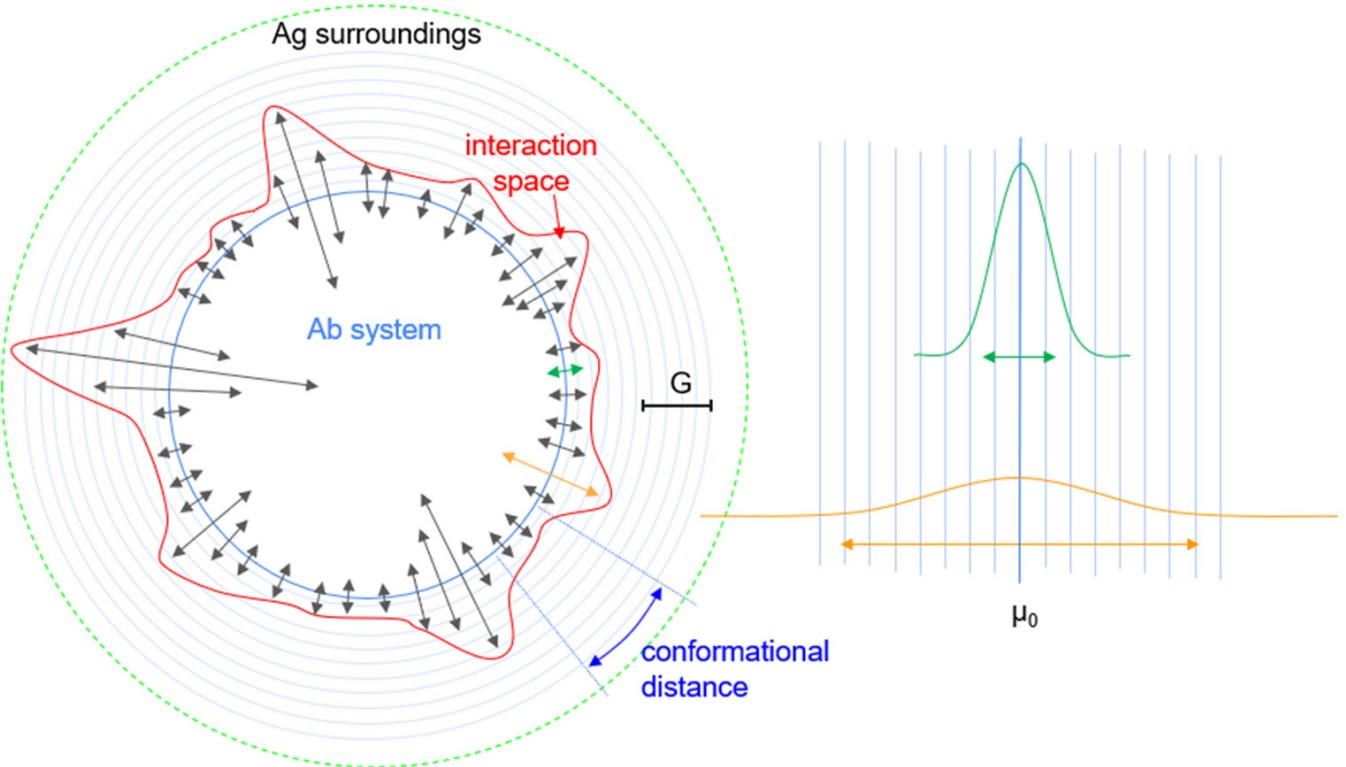

**Figure 3.** System organization by mixture of exponential and normal distributions, shown in energy scale. Green and yellow double arrows correspond to the green and yellow normal distributions sampled from the interaction space with exponentially distributed variance. The self-organizing system (Ab system) comprises units (Abs) capable of interacting with the surroundings (Ag surroundings). The free energy of the interactions has normal distribution on a system level. Organization is the adjustment of variance of interaction energy in response to thermodynamic activity of Ag so as to maintain the system. G, intensive free energy.

The probability density function of absolute thermodynamic activity $\lambda$ of elements of the system, as determined by chemical potentials $\mu_{Ab} \sim \ln(\lambda_{Ab})$, can be modeled by a mixture of exponential and lognormal distributions, as proposed in the RADARS model [27]. The model is based on the assumptions that:

1.  Chemical potentials of Ags in the interaction space, corresponding to the molecular free energy of the interactions of the system with the surroundings, are distributed exponentially, similar to a Boltzmann distribution;
2.  Equilibrium binding constants of Abs are distributed lognormally [28], with a corresponding normal distribution of molar free energies of binding;
3.  The immune system arranges and adjusts Ab chemical potentials to maintain a steady state.

The mixture of exponential and lognormal distributions, which we use here with the following parametrization, was described by Reed [29] and Mitzenmacher [30] as:

$$p(\lambda) = \int_0^\infty \alpha e^{-\alpha\mu} \frac{1}{\sqrt{2\pi\mu}} \frac{1}{\lambda} e^{-\frac{1}{2}\frac{(ln\lambda)^2}{\mu}} d\mu \tag{1}$$

where:

$\alpha$ is the rate of the exponential distribution of chemical potential of Ag;

$\lambda$ is the absolute thermodynamic activity of Abs;

$\mu$ is the chemical potential of Ag, the distance from average chemical potential $\mu_0$.

Solving the integral (see Appendix A), the result is a power law function for $\lambda \leq 1$, that is, for the system core,

$$p(\lambda_{Ab}) = \sqrt{\alpha/2}\lambda_{Ab}^{+\sqrt{2\alpha}-1} \tag{2}$$

Whereas, for $\lambda \geq 1$, that is, in the interaction space,

$$p(\lambda_{Ab}) = \sqrt{\alpha/2}\lambda_{Ab}^{-\sqrt{2\alpha}-1} \tag{3}$$

These functions are the probability densities of the absolute thermodynamic activity of Abs in the system and also define a network of interaction pathways. Thermodynamic activity corresponds to the degree of nodes in the sense that it represents the probability of interactions. The exponent of network node degree distribution, $-(\sqrt{2\alpha}+1)$, is thus determined by the rate of the exponential distribution of the chemical potential of Ag molecules, $\alpha$, generating the interaction network.

### 3.4. Architecture of Interaction Space: A Hierarchical, Scale-Free Network

Interaction space spans the range of interaction energies between $\mu_0$ and the upper limit of non-covalent binding energy. This is the space where effector Abs establish a network of interactions for Ag transport. For the description of interaction space, we can use another mathematical approach, the combination of exponentials, as suggested by Reed [31] and Newman [32]. Reed proposed that, when observing exponential growth after an exponentially distributed time, the process exhibits power law behavior. Here, we consider the exponential relationship between Ag chemical potential and the absolute thermodynamic activity of Abs combined with the exponential distribution of the chemical potential of Ags. In terms of molecular interactions, the former is the relationship between the free energy of the molecule and the number of different states of binding. Since a greater surface area, or more non-covalent bonds available for binding (greater buried surface area in the bound molecule), corresponds to greater free energy, absolute thermodynamic activity is a summation of the number of ways of interaction. For a given molecule, therefore, it is the number of links to other molecular structures with shared binding properties, an expression of similarity in reactivity. The combination of exponentials with the generation of networks can be illustrated by the overlay of activity maps (Figure 4) and the mapping of the pathways of Ag flow.

Again, we assume that the chemical potential $\mu$ of Ags is distributed exponentially when the immune system is in a resting, steady state.

$$p(\mu_{Ag}) = \alpha e^{-\alpha\mu_{Ag}} \tag{4}$$

If the absolute thermodynamic activity $\lambda$ of Abs is related to Ag chemical potential as

$$\lambda_{Ab} = \beta e^{\beta\mu_{Ag}} \tag{5}$$

then the combination of these exponential functions, the distribution of Ag chemical potential and the thermodynamic activity of Abs is given by

$$p(\lambda_{Ab}) = p(\mu_{Ag})\frac{d\mu_{Ag}}{d\lambda_{Ab}} = \frac{\alpha e^{-\alpha\mu_{Ag}}}{\beta e^{\beta\mu_{Ag}}} = \frac{\alpha}{\beta}\lambda_{Ab}^{-1-\frac{\alpha\mu}{\beta\mu}} = \frac{\alpha}{\beta}\lambda_{Ab}^{-1-\frac{\alpha}{\beta}} \qquad (6)$$

In other words, the combination of the exponential distribution of the chemical potential of Ag and the exponential relationship between Ag chemical potential and Ab thermodynamic activity determines properties of the energy transduction network and is defined by the power law distribution of Ab thermodynamic activity. Of note, this expression is an alternative to the system equilibrium binding constant $K_{sys}$ used in a previous publication [27]. The value of $\lambda_{Ab}$ reflects the number of pathways a single bond can evolve into a full binding surface and is, therefore, the number of links to nodes lower in hierarchy. Absolute thermodynamic activity thus corresponds to node degree, and its distribution determines network node degree distribution in the system.

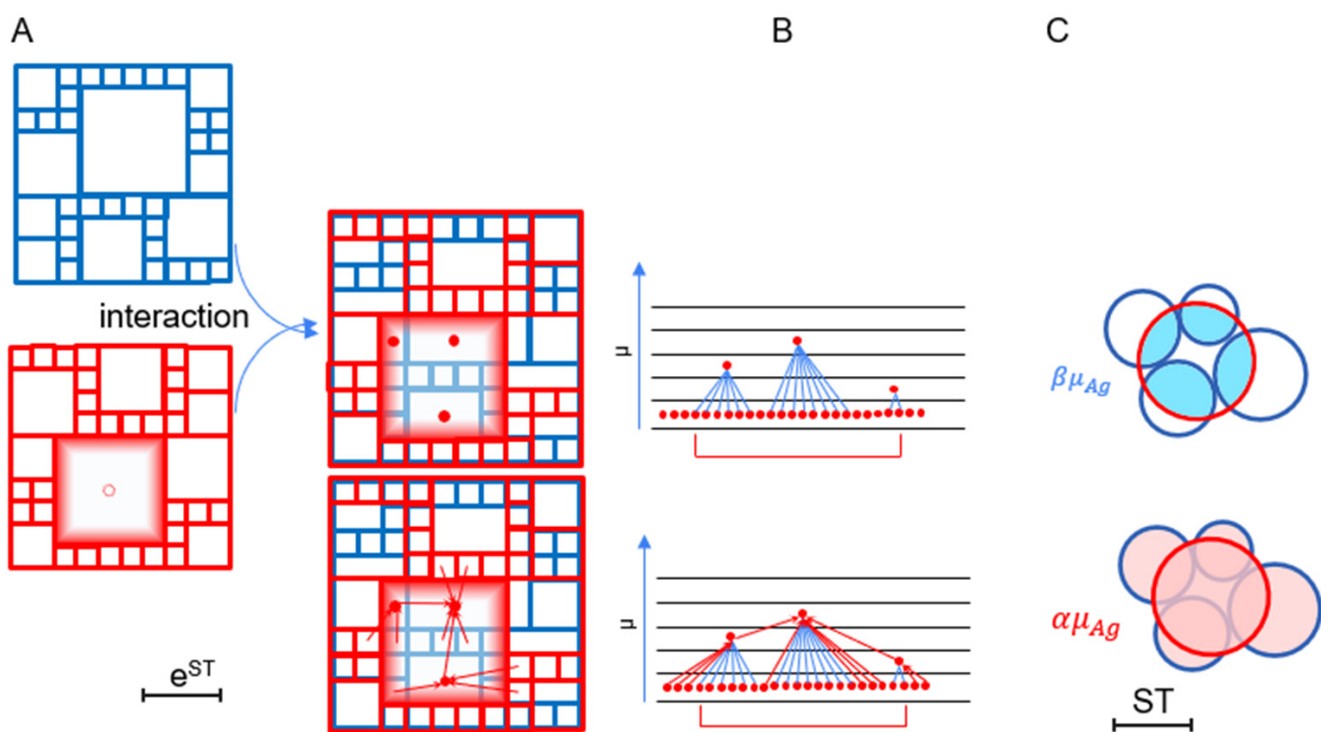

**Figure 4.** Schematic organization of antigen transfer networks and network hierarchy of molecular interactions. (**A**) By first overlaying the activity maps of antigens (red grid) and antibodies (blue grid), we obtain network nodes. Nodes (red dots, shown only for highlighted box) are assigned to each box in the overlayed maps. (**B**) Nodes are sequentially linked within each red box by applying grids with increasing side length and joining nodes in smaller boxes to nodes in the biggest box until all nodes with the red box are linked. Node hierarchy resulting from own links (blue) and renormalization links (red) are shown in energy diagrams for the shaded red box. (**C**) Minimal (upper diagrams) and maximal (lower diagrams) binding energies associated with the network nodes are characterized by β and α values, respectively, and represent intersections and the unions of conformational entropy of Ab–Ag interactions.

Beyond relating Ab and Ag thermodynamic properties, the value of β influences how nodes are linked during renormalization. Increasing $\alpha/\beta$ is accompanied by an increasing hierarchy of sub-nodes (Figure 4). From Equation (6), we can see that, in order to keep the value of degree exponent between 2 and 3, the scale-free network regimen, $\alpha/\beta$ should be in the range $1 < \alpha/\beta < 2$. Comparing Equation (6) and Equation (3), we can see that

$$p(\lambda_{Ab}) = \sqrt{2\alpha}\lambda_{Ab}^{-\sqrt{2\alpha}-1} = \frac{\alpha}{\beta}\lambda_{Ab}^{-1-\frac{\alpha}{\beta}} \tag{7}$$

Therefore, $\beta = \sqrt{\alpha/2}$ and

$$\frac{\alpha}{\beta} = \sqrt{2\alpha} \tag{8}$$

meaning that the assumption of normally distributed energies with a variance of 1 restricts the possible values of β, and the value of α alone, via determining the distribution of Ag chemical potentials, determines the distribution of Ab thermodynamic activity and the degree distribution of the Ag transport network.

*3.5. Distribution of Energy by Links in the System*

The humoral adaptive immune system is a transport system for Ag molecules. The flow of Ag molecules in the interaction space of the system, resulting from a concatenated series of interactions, is an energy transduction process. As an Ag molecule is transferred from a lower-affinity Ab to a higher-affinity Ab, the free energy of the system decreases. We may examine in the system the pathways of energy transduction by assigning direction and weight to links. The direction of Ag transport is from lower to higher chemical potential, so links are always directed to nodes with a higher degree. We can assign to each link the chemical potential difference between the two nodes connected by the link. The molecular basis of this assignment is the capability of Abs to take over Ag from weaker binders (lower chemical potential). As Figure 4 suggests, renormalization results in the joining of links with lower weight to higher-energy nodes, with hierarchy being inversely related to link weight. The number of links assigned to a chemical potential energy level is the sum of in-degrees of the nodes of that level. The average number of interchangeable links to nodes of a given chemical potential can be obtained from the relative number of direct links β and the rate variable of the distribution of Ag chemical potentials α as

$$g\mu = \frac{\beta}{\alpha} e^{\frac{\beta}{\alpha}\mu_{Ab}} \tag{9}$$

This value is the degeneracy $g$ of the energy level. Links of identical chemical potential leading to the same node are interchangeable; they represent degeneracy in binding states. The total number of accessible states or molecular partition function Z is given by the integral of the product of Equation (9) and the exponential distribution of Ab chemical potential obtained from the distribution of thermodynamic activity in Equation (6) as

$$Z = \int Z_\mu d\mu = \int g_\mu p(\mu_{Ab})d\mu = \int e^{-(\frac{\alpha}{\beta}-\frac{\beta}{\alpha})\mu_{Ab}}d\mu \tag{10}$$

In other words, with increasing node chemical potential, the increase in activity of the nodes is limited by the number and weight of incoming links, as determined by the hierarchy of sharing interaction energies. The logarithm of the weighted probabilities of interactions in the energy shells, of the number of microstates the system ranges over, is a thermodynamic potential, entropy (S). For a system consisting of N elements, the entropy can be obtained from the partition function as

$$S = \frac{U}{T} + Nk_B \ln Z \tag{11}$$

where U is internal energy, T is thermodynamic temperature, $k_B$ is the Boltzmann constant and N is the number of particles in the system. An ideal state of the system is described by $(\beta/\alpha)\mu - (\alpha/\beta)\mu = -\mu$, that is, when $\alpha/\beta = 1.618\ldots$ , the golden ratio (19). The number of accessible states, $Z_\mu$, as a function of Ab chemical potential in this configuration, is given by

$$Z_\mu = e^{-\mu_{Ab}} \tag{12}$$

and describes energy transport in an ideal, stationary state, supported by a golden network with an optimal balance between increasing node energy and the sharing of transport.

*3.6. Thermodynamic Validation of the Model*

The mathematical constructs obtained from the model so far can be corroborated by understanding the physical meaning of the variables $\alpha$ and $\beta$. Zheng and Wang proposed [28] that the distribution of the equilibrium binding constant is related to the conformational space available at the given temperature and thermal fluctuation, which is, in turn, related to the heat capacity and flexibility of the molecules [33–35]. A more flexible interaction allows for the binding of Ag to distinct Ab clonotypes, exploring greater conformation surface, while a rigid Ag epitope shows stronger preference for a single clonotype. On the other hand, distinct, flexible Ab clonotypes may bind to the same Ag, their probability-weighted chemical potentials determining the average binding energy of Ag. A higher value of $\alpha/\beta$ thus corresponds to greater flexibility and binding promiscuity, and the tail of the distribution becomes flatter; the probability of high-affinity interactions in a steady state decreases (Figure 5).

In analogy to the heat capacity ratio, which relates heat capacity at constant pressure to heat capacity at constant volume,

$$\frac{c_P}{c_V} = \frac{\left(\frac{\partial U}{\partial T}\right)_P}{\left(\frac{\partial U}{\partial T}\right)_V} = \frac{T\left(\frac{\partial S}{\partial T}\right)_P}{T\left(\frac{\partial S}{\partial T}\right)_V} \tag{13}$$

The ratio of $\alpha/\beta$ defines a binding flexibility ratio in configuration space where

$$\frac{\alpha}{\beta} = \frac{T\left(\frac{\partial S}{\partial T}\right)_\mu}{T\left(\frac{\partial S}{\partial T}\right)_N} \tag{14}$$

This relates binding flexibility to chemical potential $\mu$, kept constant to flexibility with number of system elements N kept constant, in the following sense: The partial differential expression in the numerator describes the extent of conformational surface S explored by Ag molecules when these molecules are provided by the surroundings so as to keep chemical potential $\mu$ constant. The expression in the denominator describes the potential to explore the conformational surface due to thermal fluctuation T; when no extra molecules are supplied, N is invariant. The immune system is, indeed, capable of fine thermodynamic tuning of the interactions of its Abs by selecting the structure with appropriate flexibility by isotype switching and affinity maturation and by targeting selected Ag epitopes. The regulation of these Ab properties for each and every Ag is the essence of controlling Ag concentrations.

If we regard blood plasma as an open, single-phase, multicomponent system, we can use a thermodynamic potential for the description of energies related to Ab–Ag interactions. Following the nomenclature used by Emmerich [36] for a thermodynamic potential expressed for extensive entropy,

$$\Lambda = S + \sum_i \frac{\mu_i}{T} N_i \tag{15}$$

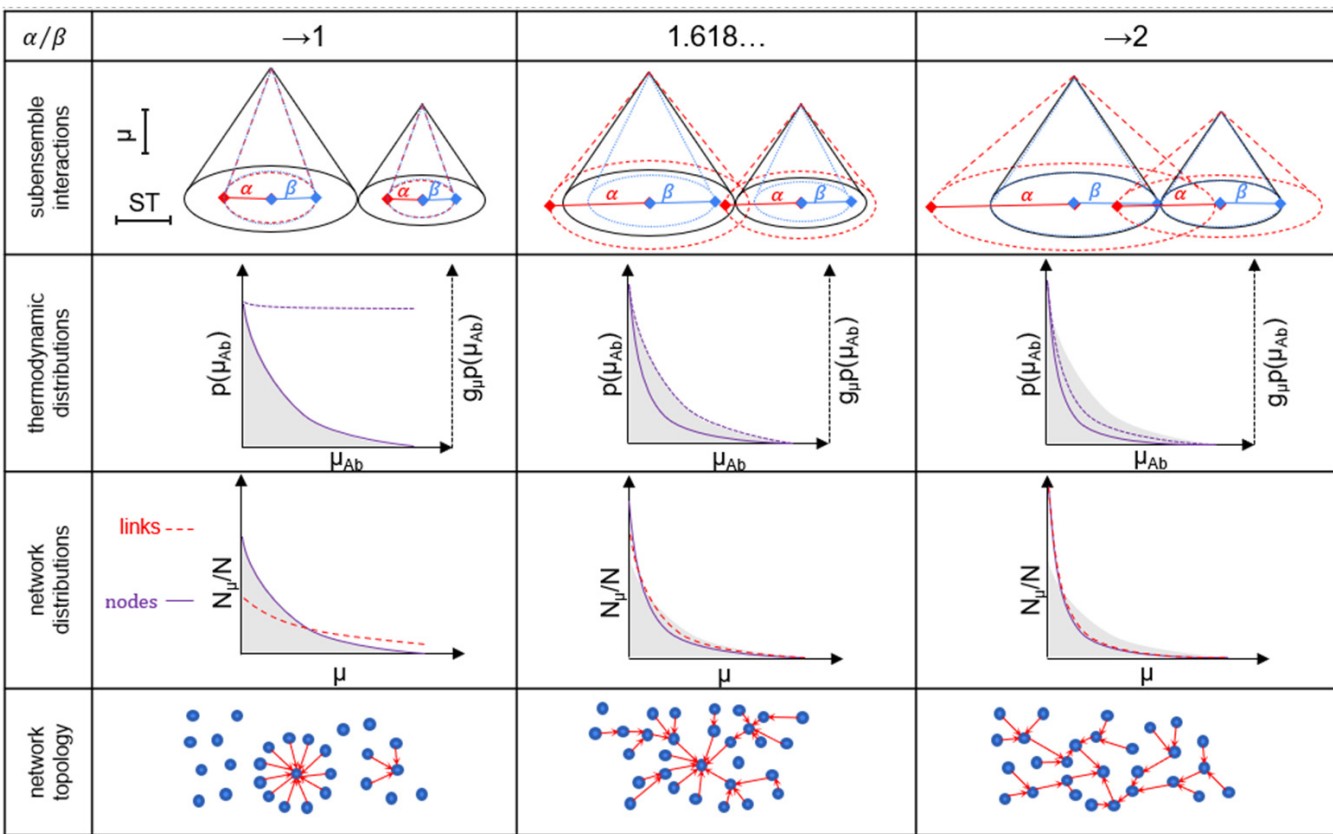

**Figure 5.** System properties as determined by the binding flexibility ratio $\alpha/\beta$. The relative arrangement of two binding ensembles, here represented as cones, in interaction space is shown. The effects of increasing the value of the ratio, exerted via increasing $\alpha$ (red) and $\beta$ (blue), on the coverage of the conformational entropy surface is depicted by the base of cones and their overlap. The distribution of chemical potentials (solid line) and of partition function (dashed line) is shown against a reference exponential distribution (filled gray) with exponent $-1$. The distribution of network node chemical potentials, corresponding to Ab chemical potentials (solid purple), and of link chemical potentials, corresponding to Ag chemical potentials (dashed red), for different values of the ratio are shown against the reference exponential distribution in gray. The bottom row is the schematic topology of networks for the three different values of $\alpha/\beta$ ratio shown at the top. The relationship between $\alpha$ and $\beta$ is defined by Equation (8). $p(\mu_{Ab})$, probability density of Ab chemical potential; $g_\mu p(\mu_{Ab})$, product of degeneracy and probability; $N_\mu$, number of nodes or links with given chemical potential; N, total number of nodes or links; $\mu$, chemical potential.

Thermodynamic potential $\Lambda$ summarizes the essence of the adaptive thermodynamic system: create a large conformational landscape (conformational entropy S) and generate N elements with chemical potential $\mu$ expressed as the ability to cover conformation space,

$$\frac{\partial S}{\partial N} = -\frac{\mu}{T} \tag{16}$$

Equation (16), in our model, states that the addition of molecules to the system changes the conformational entropy according to the molecular free energy $\mu$ per unit temperature T. We can replace the summation in Equation (15) with the probability-weighted sum of chemical potentials, which is the average chemical potential $\langle\mu\rangle$, and combine with Equation (11) in the form:

$$\Lambda = N\left(k_B lnZ + \langle\frac{\mu}{T}\rangle\right) \tag{17}$$

Thus, the thermodynamic potential Λ of Abs depends on their number *N*, the number of interactions they can engage in, summarized in *Z*, and their average chemical potential. The configuration space model introduced here arranges constituents of the system based on their "power" to change other constituents' chemical potential. This power is reflected in the topology and network linkage of constituents. In a stationary state, the distribution, and, thereby, the average chemical potential of all Ag molecules, is kept constant, adjusting binding energy that tickles but does not activate sensors, memory or naïve B cells. This is achieved by setting a chemical potential that immunologically suits the host: low chemical potential for harmless self molecules, high chemical potential for dangerous non-self molecules. Concentrations of targets are then adjusted accordingly, meaning that Ags with high chemical potential are eliminated with higher efficiency. The process leads to a wide range of chemical potentials of Ags and Abs in the system. A theoretical, system-wide steady state is reached when Ab free energies are distributed exponentially. In the ideal configuration of Equation (12), the partition function Z is equal to one; thus, Equation (17) simplifies to the product of element number N and the average chemical potential.

Self-organization of the system refers to how many Ab molecules are generated and how Ab chemical potentials are distributed. In a thermodynamic steady state, the system minimizes free energy and maximizes entropy; this state can be described by statistical mechanics functions. The adaptive, self-organizing biological ensemble is a co-existing collection of large numbers of copies of states, with a regulated total extensive free energy G and a composition that may fluctuate with the distribution of chemical potentials adjusted via network formation (Table 1). In immunological terms, in order to reach a steady state, the chemical potential of Abs in the system has to be tuned with regards to all potential binding partners, not just the Ag that triggered a response. The pathway to reaching the steady state is the selection of Abs (cell clones) that fit into a previously organized network. Steady state is reached when the system, again, possesses an extensive free energy G that continuously drives the binding of Abs to Ags.

**Table 1.** Comparison of physical ensembles.

| Ensemble | Constant [1] | Adjusted |
|---|---|---|
| microcanonical | E | microstates |
| canonical | T | E |
| grand canonical | T, μ | E, N |
| adaptive biological | $G/T = \sum N\mu/T$ | μ/T, N |

[1] Pressure is assumed constant in all these ensembles.

The biological, self-organizing system diversifies its tool of energy transfer, Ab molecules in our case, adjusting the distribution of chemical potentials so as to maintain the system against forces of change in the environment. By deploying mechanisms to sense the free energy of constituents in the environment, the system can adjust, readjust and evolve with the environment.

*3.7. Immunological Validation of the Model*

The humoral immune system must regulate, over a very wide range, the concentration of a vast number of molecules that are found in and constitute an organism. How it is technically possible to eliminate certain molecules while leaving others unharmed has led to the long-standing question of tuning recognition specificity and affinity [37], breadth and depth [38]. The solution provided by an adaptive immune system apparently requires a scale of numerosity, diversity and affinity comparable to that found in the organism—or, more accurately, the supraorganism [39]—itself. A mechanistic approach to the feedback and tuning procedure, based on saturation of Abs and Ags, was proposed in a quantitative model of B-cell development and Ag removal [8,40]. The model treated distinct molecules independently, neglecting the effect of cross-reactivity but, nevertheless, providing a general framework for understanding the system. On the

level of individual cells, the level of engagement of Ag receptors determines the cell's fate: programmed death, survival, proliferation or differentiation. An initial repertoire of naïve cells, which are constantly generated, displays receptors produced by the random rearrangement of gene segments and subsequent selection steps. The immunologically controlled encounters of these cells with the Ags of the inner and outer environment lead to the expansion and differentiation of clones according to the immunological ranking of the target Ag. These processes are happening continuously, manifesting as the adaptation of the system to the antigenic environment or, in other words, as self-organization. As a result, a repertoire of memory B cells and long-lived plasma cells is produced, which reflects past adaptation and serves as the basis of further evolution in response to the environment [20,41].

The present model identifies variables that characterize network formation and define the system on a thermodynamic basis. Both Abs and Ags are cross-reactive, being able to bind multiple partners. The average binding flexibility or cross-reactivity on the system level manifests as the slope of the distribution curves when a logarithmic scale is used (Figure 6). Changing the slope of either influences the other via self-organization, as described above. Complexes of Abs and Ags are constantly removed from the system via cells with Ab Fc receptors. Changes in the composition and availability of Ags trigger the activation of sensor cells, leading to readjustment of the properties of the repertoire (Figure 6). It is tempting to speculate that the local hierarchy of binding has important immunological consequences; minimal hierarchy would result in a "powerful" Ab response, in the sense of high-affinity binding and appropriately switched Ig isotypes, such as an allergic IgE response. With the application of novel tools for the deeper analysis of serological reactions, such questions can be experimentally addressed [18].

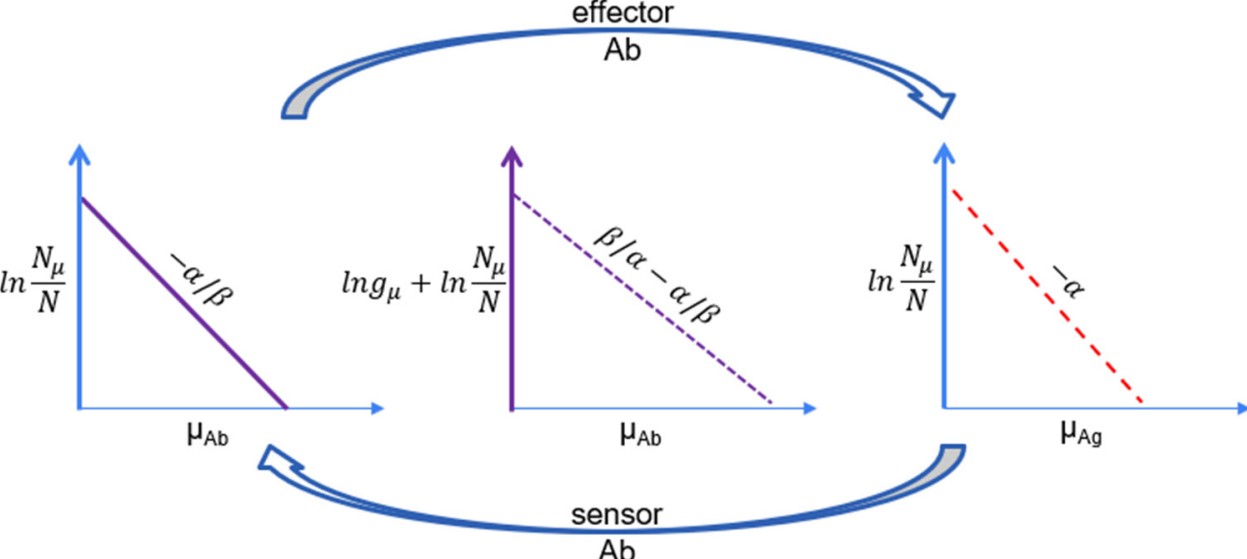

**Figure 6.** Immunological aspects of the distributions. The exponential distribution of chemical potentials of both the Abs and Ags are maintained by the feedback loop of effector and sensor Abs. On a logarithmic scale, the slope of the distributions is indicated for Ab chemical potentials (solid purple), Ab–Ag complexes (dashed purple line) and Ag (red). The immune system reduces the concentration of Ags according to its chemical potential that is adjusted by Abs and removes Ab–Ag complexes to maintain Ag flow. $N_\mu$, number of events with chemical potential $\mu$; N, total number of events; $g_\mu$, degeneracy of events with chemical potential $\mu$.

Immunoassays probe the system of Abs via measuring Ab binding to selected Ags. These assays are also called serological assays because they characterize serum Ab reactivity against medically relevant Ag targets. An immunoassay that uses Ag titration, a gradient of Ag concentrations, can be interpreted as the measurement of the changes of

chemical potential in the interaction space and can be modeled by the Richards differential equation [20]. In the sigmoid Richards curve, the point of inflection corresponds to the sum of probability-weighted chemical potentials and is, therefore, a measure of Ab affinity. Meanwhile, the asymmetry parameter of the Richards function is related to $\alpha/\beta$ in the system, which corresponds to the limiting activity coefficient of Ags. Thus, the variable that describes the hierarchy and network organization of Abs in the system appears in a biochemical measurement as a thermodynamic variable.

A theoretically important message of the model is the continuity of self. The interaction space belongs to the system; it is self. However, it also incorporates elements of the surroundings in a regulated way. Ag-binding energies in the interaction space range from system average to very far from average, and elements are distributed according to an exponential distribution. Therefore, the frequency or mole fraction contribution to the system also covers a very wide range (Figure 7). All binding events belong to the system, becoming incorporated into the architecture as imprints in the network. This is consistent with the liquid hypothesis of self, which states that immunological identity is continuous and dynamic [42]. Even though we have no exact physical models describing the functioning of the immune system, we have more and more experimental data on the structural and molecular mechanisms of Ab function and networks of Abs based on sequencing. It is, thus, possible to relate our model to these observations. The total immunoglobulin concentration ($[Ab]_t$) in adult human plasma is in the high micromolar range (~$10^{-4}$ M). We can express Ab quantities as a fraction of the system by using this reference as $[Ab]/[Ab]_t$ (Figure 7).

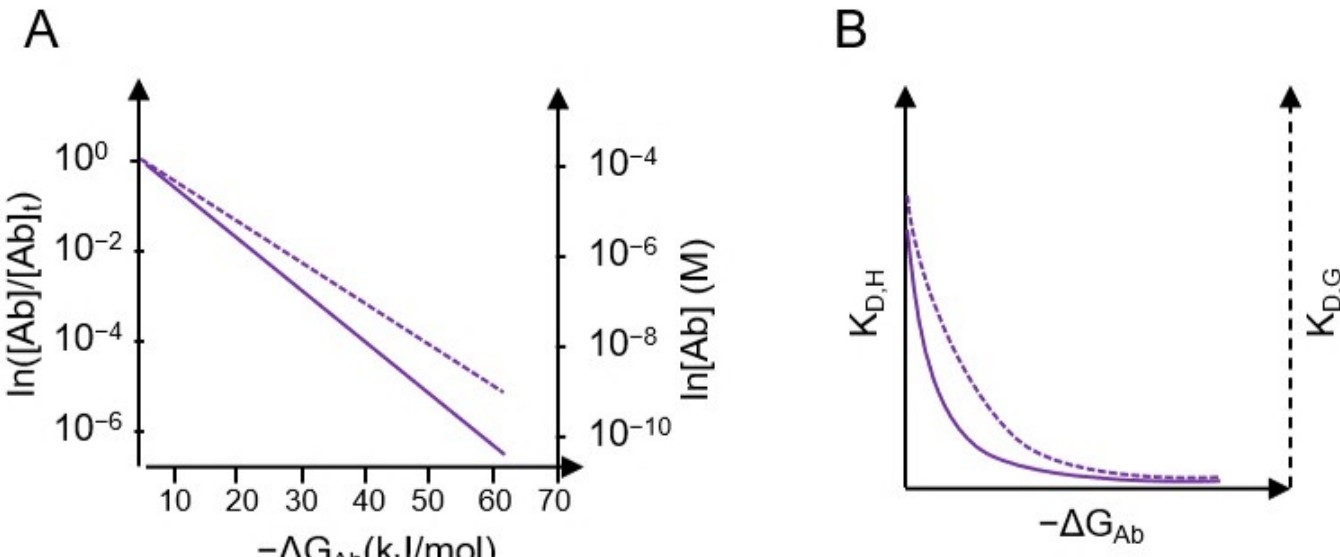

**Figure 7.** Antibody concentrations and equilibrium affinity constants. (**A**) Relative (left axis) and absolute (right axis) concentrations of Abs with indicated binding energies. (**B**) $K_D$ values corresponding to binding enthalpy for a monoclonal Ab solution (solid purple line) and for a serum Ab (dashed line) with entropy effects. $\Delta G_{Ab}$, free energy of interaction; $K_{D,H}$, equilibrium dissociation constant calculated from enthalpy; $K_{D,H}$, equilibrium dissociation constant calculated from Gibbs energy.

Assuming that the system is in a steady state when all its components are present at their equilibrium dissociation concentrations $K_D$, their distribution as a function of chemical potential is determined by the standard free energy of their interactions. Gibbs free energy has two components:

$$K_D = e^{\frac{\Delta_r G^\circ}{RT}} = e^{-\frac{\Delta_r S^\circ T}{RT}} \times e^{\frac{\Delta_r H^\circ}{RT}} \tag{18}$$

Namely, an entropic (S) and an enthalpic (H) component. We can relate the free energy of binding to these components using the relationship between the distributions of chemical potentials and interactions, combining Equations (10) and (18) as

$$K_D = e^{-\frac{\beta}{\alpha} \frac{\Delta_r G^\circ}{RT}} \times e^{\frac{\alpha}{\beta} \frac{\Delta_r G^\circ}{RT}} \tag{19}$$

The difference between using only the enthalpic component or both components is illustrated by the solid and dashed lines, respectively, of Figure 7. The concentration of an Ab clone with a particular binding energy in the system in steady state is higher than that of the same Ab alone in equilibrium with its single binding partner. Simply stated, because interaction energies are "spread out" over the network, equilibrium concentrations in a system are higher than in a solution of two components, such as a monoclonal Ab and a single target Ag.

*3.8. Network Validation of the Model*

A theoretical network of Abs can be generated on the basis of sequence and structure similarity [27], which corresponds to experimentally determined, Ag-binding correlation networks [43]. On the molecular level, these networks allow the transfer of Ags from an Ab node to another Ab node in the direction of increasing affinity. The configuration space model of this article organizes such links into a hierarchy of chemical potential and renders structural similarity into topological relationships. Increasing chemical potential in the configuration space means shifting in the hierarchy and increasing node degree, which is related to the probability of receiving an Ag molecule from a node lower in hierarchy. All nodes high in the hierarchy are "supported" by a large number of nodes with lower chemical potential. This arrangement causes the spreading of hubs over the interaction space and manifests as the repulsion of hubs and results in a disassortative network.

The power law degree distribution and scale-independent property of the network were previously derived from a lognormal distribution of equilibrium binding constants of random interactions in combination with an exponential distribution of the standard deviation of binding energy [27]. Here, a power law distribution is also obtained from a Boltzmannian distribution of molar free energy of Ags and the relationship between the absolute thermodynamic activity of Abs and the chemical potential of Ag. Cross-reactivity is the result of partial interactions with lower-than-maximal energy. Higher-energy inter-actions allow more cross-reactivity, since a greater molecular surface area is available for binding. This cross-reactivity appears in the renormalization strategy (Figure 4), where nodes with lower degree are joined to nodes with a higher degree. This renormalization is similar to the box-counting method of determining fractal dimension [44] in the sense that grids with increasing side lengths are applied to reveal network architecture. Disassortativity and fractality have been recognized as features of biological networks [45,46].

From a network perspective, the ratio of our variables $\alpha/\beta$ corresponds to the ratio of fractal dimension $d_B$ and degree exponent of boxes dk, as described by Song et al. [45,47]. The network degree distribution exponent $\gamma$ can be calculated from these as

$$\gamma = 1 + \frac{d_B}{d_k} = 1 + \frac{\alpha}{\beta} \tag{20}$$

This is in agreement with our findings that $\alpha/\beta$ is the rate parameter of the exponential distribution of chemical potential, and $\alpha/\beta+1$ is the exponent of the power law distribution of thermodynamic activity and network degree distribution (Equation (6)) (also see Appendix B). The evolution of the network in time is reflected in configuration space, in as much as the nodes with greater chemical potential that develop in the time course of immunological reactions are located more distantly from the system surface. Therefore, these indices can be interpreted as factors of renormalization in energy levels.

According to Caetano-Anollés et al. [48], scale-free networks can follow a trajectory in network morphospace from homogenous, non-modular towards heterogenous, modular

network organization. This trajectory can correspond to a mesh-like structure of random binding events at the core surface of the system, evolving into a highly hierarchical and modular network of specific and high-affinity interactions. This corresponds to a star-like network distribution of energy when $\alpha/\beta$ approaches 1 and multiple transfers of energy when $\alpha/\beta$ increases (Figure 5). Whereas the probability density of binding energy is associated with a Boltzmannian distribution, scale-free networks are known for the power law distribution of network node degree. Our configuration space model of a system suggests that these two phenomena are two sides of the same coin; an identical, complementary cumulative distribution results in the exponential probability density function of chemical potential and power law probability function of node degrees of elements of the system (see Appendix B).

### 3.9. Interpretation of the Model as a Complex System

In their maximum-entropy model for Ab diversity, Mora et al. associated Ab sequences with effective energies as if the sequences represented a particular state in a system in equilibrium [16]. In this article, it is suggested that Ab structures, which are defined by their amino acid sequences, are, indeed, distinct states in the configuration space of the system. The molecular free energy of these structures is determined by the composition of the system, which is itself regulated by the system.

Following the active, expansive stage of the immune response, a contraction stage establishes a balance between the thermodynamic activity of Abs and of Ags by selecting only a subset of the cells that evolved in the active stage. This process leads to a steady state, whereby secreted Abs are continuously removed along with bound Ags, keeping both free Ab and free Ag concentrations at the immunologically adjusted values. A two-way feedback mechanism of sensor and effector cells adjusts chemical potential so that cells are poised between under- and overactivation (see [8,20,40] for details). The effector cell secretes Abs capable of reducing target Ag concentration; the extent of reduction is determined by the chemical potential of the Ab. The sensor cell receives an amount of energy in the form of survival signals by the target Ag—this is called tickling in immunological jargon. This signal is adjusted by the effector cells, secreting soluble Abs. From an immunological point of view, this means the parallel generation of memory B cells (sensors) and plasma cells (effectors) in germinal centers with closely related if not identical binding properties [49–51]. Perturbations at any point in the interaction space trigger the rearrangement of this hierarchy and network. In this sense, the system is poised at criticality, a phenomenon suggested to be present in all biological systems [52]. Critical events represent the reorganization of hierarchy in a system. These events are the coalescence of sub-ensembles or subnetworks. The size of these events often shows power law distribution over time (strength of earthquakes, size of forest fires, avalanches of sandpiles) [53]. The immune system presumably reorganizes itself constantly, adapting to the antigenic environment, via such events. Occasionally, triggered by infections, massive reorganization is necessary, which may correspond to a huge critical event.

A variety of phenomena has been shown to follow power law, including natural events and systems and artificial, man-made systems. In these systems, the power law applies only for values greater than minimum value $x_{min}$. The distributions described here suggest that the minimum value of x corresponds to activity with the average energy or reference chemical potential $\mu_0$ of the system. By our definition, this reference value is zero, and the corresponding activity is one.

$$\lambda_{min} = e^{\mu_0} = 1 \tag{21}$$

It also means that, in our directed network, all isolated nodes possess an in-degree of 1 (see Figure 5). Nodes with a higher degree connect to form the networks, which represent the organization of the interactions of the system with its environment.

Power law relationships can be broadly assigned into two main categories: distributions of frequencies and of magnitudes [54]. Complementary cumulative distributions (cCDF) of frequencies, rank–frequency plots, Zipf's law and the Pareto distributions are

examples of the first type [30,55]. Protein interaction and metabolic networks are examples of the second type [56,57].

Magnitudes represent an intensive physical property of the adaptive system. The distribution of an intensive physical property describes a hierarchy and often represents a hierarchical network that organizes the system. It follows that these magnitudes are related to the network degree of the entities. The configuration space model presented here suggests that these magnitudes are absolute thermodynamic activities and are related to molar free energy or chemical potential, an intensive physical property.

Frequency is an extensive property and corresponds to the number of links (interactions) of a given energy in our model. If the energy of interactions is distributed exponentially (Equations (4) and (12)), then the logarithm of probability (Equation (9)) decreases linearly (Figures 6–8). The corresponding linear growth of entropy (for cumulative distribution) is characteristic of criticality in thermodynamic systems [16,58]. In unchanging, simple surroundings, the system can strongly adapt (Figure 8, vertical arrow) and distribute its energies over a wider range of chemical potentials. On the other hand, changing, diverse surroundings prohibit effective adaptation (Figure 8, horizontal arrow), allowing large numbers of weak interactions. The system can develop and grow, maintaining its organization by parallel growth of its intensive and extensive properties, chemical potentials and the number of interactions as a function of those potentials, respectively.

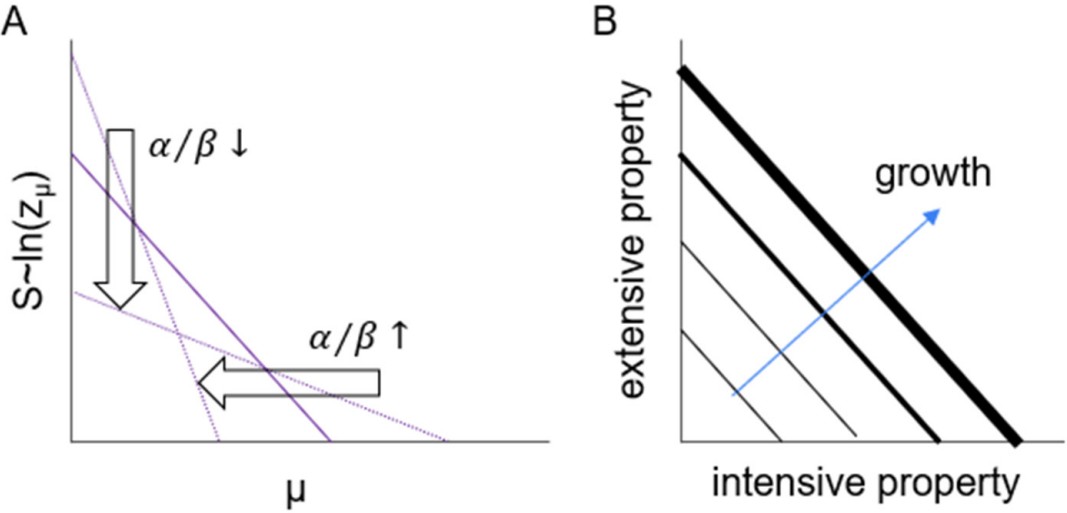

**Figure 8.** Graphical representation of relationships of system variables. (**A**) The relationship between entropy and chemical potential of system elements results from the balanced distribution of interaction energies and accessible binding states. Too strong an adaptation ($\alpha/\beta$ decrease) or too weak an adaptation ($\alpha/\beta$ increase) results in less stable configurations (dashed lines). (**B**) Balanced growth maintains organization by maintaining relations between these intensive and extensive properties. S, entropy; $z_\mu$, partition function of energy level $\mu$.

Power law distributions, in general, characterize critical points and phase transition. In this respect, self-organization aims to maintain the state of phase transition. In an abstract sense, transition is between the two phases of organization of matter in the system and in the surroundings. Self-organization thus counteracts the effects of the surrounding environment by generating an interaction space and, therein, sustaining phase transition, maintaining steady-state reactions towards all environmental components that would otherwise disintegrate the system.

## 4. Conclusions

There can be little doubt that a system with the number of constituents and the extent of diversity that the adaptive immune system possesses can behave as a complex, adaptive physical system. The more important question is, perhaps: would a complex biological

system follow the rules that apply to a thermodynamic system? The thermodynamic system within the complex biological system of immunity can be defined as an open, multicomponent system of molecular interactions between antibodies and antigens. As compared to a physical system with very clear boundaries, e.g., a thermally insulated bottle with gas molecules filling it, the biological system can be thought of as treating only a part of the biological system as a thermodynamic system. Thus, it has no strict physical boundaries; antibodies and antigens interact everywhere in the organism, with some privileged sites, such as the blood plasma, the lymphoid organs and the mucosal surfaces. The boundary is present only in the abstract, theoretical space because we consider only these interactions. Because of this abstraction, and because this system is embedded in a highly complex and dynamic environment, the model only serves to obtain the general laws and global variables of the system. This is why we regard the system to be an open one. Unlike in the case of the Thermos bottle or chemical reactions taking place in a test tube, no precise calculations can be made for each of the interacting components. Yet, the validity of the model can be supported by the analogies described in the paper and by the experimental technologies that examined a singular antigen's reactivities. Of the multicomponent system, selected subsets of the components can be measured by serological assays.

This article identified attributes of the humoral adaptive immune system that seem to capture the physics of the biological system. These attributes are the magnitudes and frequencies of Ab chemical potentials. A coefficient that appears in all approaches to describing the system is the heat capacity ratio, or binding flexibility ratio or degree distribution exponent. Beyond the theoretical advancement in modeling, the introduced mathematical framework can also be put into practice in quantitative serological measurements [18,20] where this coefficient is the limiting thermodynamic activity coefficient and can be experimentally obtained. Such measurements could provide quantitative maps of the Ab configuration space in the future.

**Funding:** This work received no specific funding.

**Institutional Review Board Statement:** Not applicable.

**Informed Consent Statement:** Not applicable.

**Data Availability Statement:** No new data were created.

**Acknowledgments:** Special thanks to Tamás Pfeil (ELKH-ELTE Numerical Analysis and Large Networks Research Group, Budapest) for reading the manuscript and correcting my mathematical mistakes. Without his help, I could not have clarified the math behind these phenomena. Many thanks to Tamás Vicsek (ELTE, Department of Biological Physics, Budapest) for taking his time to read the manuscript and advise me on the backgrounds of this huge field of science.

**Conflicts of Interest:** The author declares no conflict of interest.

**Appendix A**

Derivation of power law from combination of exponential and lognormal functions [29,30]:

$$p(\lambda) = \int_0^\infty \alpha e^{-\alpha\mu} \frac{1}{\sqrt{2\pi\mu}} \frac{1}{\lambda} e^{-\frac{1}{2}\frac{(ln\lambda)^2}{\mu}} d\mu$$

Rearranging, after substituting $\mu = u^2$ in the integral,

$$p(\lambda) = \frac{2\alpha}{\sqrt{2\pi}} \frac{1}{\lambda} \int_{u=0}^\infty e^{-\alpha u^2 - \frac{(ln\lambda)^2}{2u^2}} du$$

Using integral the table identity,

$$\int_0^\infty e^{-az^2 - b/z^2} dz = \frac{1}{2}\sqrt{\frac{\pi}{a}} e^{-2\sqrt{ab}}$$

and substituting $\alpha$ for a and $(ln\lambda)^2/2$ for b,

$$\int_0^\infty e^{-\alpha z^2 - (ln\lambda)^2/4z^2} dz = \frac{1}{2}\sqrt{\frac{\pi}{\alpha}} e^{-2\sqrt{\alpha(ln\lambda)^2/2}}$$

Replacing the integral part of the equation,

$$p(\lambda) = \frac{2\alpha}{\sqrt{2\pi}} \frac{1}{\lambda} \frac{1}{2} \sqrt{\frac{\pi}{\alpha}} e^{-2\sqrt{\alpha(ln\lambda)^2/2}}$$

Simplifying with $\sqrt{\pi}$, 2, $2/\sqrt{2}$,

$$p(\lambda) = \frac{2\alpha}{\sqrt{2\pi}} \frac{1}{\lambda} \frac{1}{2} \sqrt{\frac{\pi}{\alpha}} e^{-2\sqrt{\alpha(ln\lambda)^2/2}}$$

for $\lambda > 1$ gives

$$p(\lambda) = \frac{\sqrt{\alpha}}{\sqrt{2}\lambda} e^{-ln\lambda\sqrt{2\alpha}}$$

Then, simplifying the exponential expression,

$$p(\lambda) = \frac{\sqrt{\alpha}}{\sqrt{2}\lambda} \lambda^{-\sqrt{2\alpha}}$$

Expressing $1/\lambda$ as power:

$$p(\lambda) = \frac{\sqrt{\alpha}}{\sqrt{2}} \lambda^{-1} \lambda^{-\sqrt{2\alpha}}$$

Uniting exponents of $\lambda$:

$$p(\lambda) = \sqrt{\frac{\alpha}{2}} \lambda^{-1-\sqrt{2\alpha}}$$

## Appendix B

*Appendix B.1 Summary of Parameter Definitions*

$\alpha$: rate parameter of exponential distribution of Ag chemical potential in interaction space:

$$p(\mu_{Ag}) = \alpha e^{-\alpha\mu_{Ag}}$$

$\beta = \frac{\alpha}{\sqrt{2\alpha}} = \sqrt{\frac{\alpha}{2}}$: relation of Ab thermodynamic activity to Ag chemical potential.

$$\lambda_{Ab} \sim e^{\beta\mu_{Ag}}$$

$\frac{\alpha}{\beta} = \sqrt{2\alpha}$: binding flexibility ratio, heat capacity ratio, fractal dimension to box degree exponent ratio, related to network degree exponent as

$$\gamma = \sqrt{2\alpha} + 1$$

*Appendix B.2 Correspondence of Exponential and Power Law Distributions*

Probability density function (PDF), cumulative distribution function (CDF) and complementary cumulative distribution function (cCDF) of node with chemical potential $\mu_{Ab}$ and of absolute thermodynamic activity $\lambda_{Ab}$, corresponding to node degree; these variables are related as

$$\lambda_{Ab} = e^{\mu_{Ab}}$$

**Table A1.** Correspondence of distributions.

| PDF | $p(\mu_{Ab}) = \sqrt{\alpha/2}\,e^{-\sqrt{2\alpha}\mu_{Ab}}$ | $p(\lambda_{Ab}) = \sqrt{\alpha/2}\,\lambda_{Ab}^{-(\sqrt{2\alpha}+1)}$ |
|---|---|---|
| CDF | $P(\mu_{Ab}) = 1 - e^{-\sqrt{2\alpha}\mu_{Ab}}$ | $P(\lambda_{Ab}) = 1 - \lambda_{Ab}^{-\sqrt{2\alpha}}$ |
| cCDF | $P^{-1}(\mu_{Ab}) = e^{-\sqrt{2\alpha}\mu_{Ab}}$ | $P^{-1}(\lambda_{Ab}) = \lambda_{Ab}^{-\sqrt{2\alpha}}$ |

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
