# Peer review of "Complex Physical Properties of an Adaptive, Self-Organizing Biological System"

_biophysica, doi:10.3390/biophysica3020015_

Round 1

Reviewer 1 Report

The paper entitled : „Complex physical properties of an adaptive, self-organizing bi- 2 ological system” prepared by József Prechl discusses the problem of immune system activity and organisation. The attempt is undertaken to define the chemico-physical model using the scale-free logic and its relations to fractal organisation. Paper presentes a very important and interesting issue with innovative aspects. Of special interest to me is the „phases of organization of matter in the system and in the surroundings”. However I have doubts as to the meaning of „system”. This term is overused in the paper – see commenst below.

The problem I faced is:

1.       The lack of clear distinction of „system”. Sometimes it is used just to protein-protein interaction (antygen-antibory). The same term is used for whole immunological network. Autor shall distinguish these two approaches. The best form it could be to divide the presentation into two parts : molecular level (protei-protein interaction and the whole immunological system treated as population. And then at the end the third part focused on linking these two approaches

2.       The lack of clear definition of self-organisation – is it the proces on the molecular level (single protein – single protein interaction) or it is the discussion of the population approach of immunological organisation.

3.       Fig.4. – no explanation of σ and β distinguished on the right side of the figure.

The attributes not taken into account are:

1.       The interpretation is limited to rigid structural forms (key and lock model disussed) which is the important limitation in respect to Ab structure. Why the binding area is doubled in the structure of Ab (two chains of initially symmetrical construction) ?

2.       TThe interpretation presented in the paper under consideration stops at the moment which shall be the initiation for next step – probably the most important for immune responce – C1q complexation.

3.       Is the model of universal character ? Is it applicable to any protein-protein complexation ?

Minor comments:

1.       The Figures are not sufficiently explained in legends. All symbols present on the figure shall be explained in legend – see Fig. 2 – only two maps on left are explained in legend to this figure. This comment is valid for all figures present in the paper.

Fig 3 – G – no explanation

General rule is to give as much information as it is necessary for the Reader to recreate the presentation. It is impossible in the case of the presented paper.

2.       Fig. 3. Self-organising „system” (Ab system) ……….- Is it just one Ab molecule and one Ag molecule – or we are taking abut the large set of Abs and Ags

„response to thermodynamic activity of Ag, so as to maintain the system” – to maintain what system ? – one protein interacting with the second one or the whole population ?

Last comment on the „system” usage „μ is the chemical potential of antigen, the distance from system average chemical potential μ0” System – just in one molecule/complex  or in the whole population of Abs and Ags.

3.       Fig 8 – the line with the G – not explained in the legend

Author Response

I wish to thanks the reviewer for considering my manuscript for the review. Below please find my responses to the remarks and criticism.

The paper entitled : „Complex physical properties of an adaptive, self-organizing bi- 2 ological system” prepared by József Prechl discusses the problem of immune system activity and organisation. The attempt is undertaken to define the chemico-physical model using the scale-free logic and its relations to fractal organisation. Paper presentes a very important and interesting issue with innovative aspects. Of special interest to me is the „phases of organization of matter in the system and in the surroundings”. However I have doubts as to the meaning of „system”. This term is overused in the paper – see commenst below.
>> Re: Considering the remarks of all three reviewers I revised the whole manuscript, with the general aim of better explaining all figures and equations, simplifying language and focusing on the key messages.

The problem I faced is:

  1. The lack of clear distinction of „system”. Sometimes it is used just to protein-protein interaction (antygen-antibory). The same term is used for whole immunological network. Autor shall distinguish these two approaches. The best form it could be to divide the presentation into two parts : molecular level (protei-protein interaction and the whole immunological system treated as population. And then at the end the third part focused on linking these two approaches
    >>Re: In the manuscript I use the term “system” for all the antibody and antigen molecules and their interactions. In the revised text I tried to be consistent with this usage and this is stated explicitly in line 114. On the other hand, it is true that this system is created by the quality and strength of the molecular interactions which are assigned a coordinate in the configuration space. Therefore, I would prefer to keep the present logic in the organization of the manuscript, but I changed the section titles to better reflect this approach.
  2. The lack of clear definition of self-organisation – is it the proces on the molecular level (single protein – single protein interaction) or it is the discussion of the population approach of immunological organisation.
    >> Re: A complete section (2.2) is dedicated to the explanation of self-organization. Again, the mechanism of organization is immunological but it affects the molecules, since it is the molecular evolution (hypermutation and affinity maturation) that achieves organization in the system. I hope that the revised version of the manuscript conveys this message better than the first version.
  3. Fig.4. – no explanation of σ and β distinguished on the right side of the figure.
    >> Re: Figure 4 and all other figures are better explained in the legends in the revised manuscript, including the mentioned Greek letters.

The attributes not taken into account are:

  1. The interpretation is limited to rigid structural forms (key and lock model disussed) which is the important limitation in respect to Ab structure. Why the binding area is doubled in the structure of Ab (two chains of initially symmetrical construction) ?
    >>Re: This paper focuses on the affinity of the binding sites, the distribution of binding energies of single binding sites. The immune system uses other measures, such as isotype switching, to further modify the efficiency of binding and antigen removal, but that is beyond the scope of this paper. I do mention the fact that isotype switch (leading to heavy chains with different Ag binding valencies) is also part of the immunological arsenal, but I cannot link the lines of thought in the paper with the fact that most antibodies are bivalent, with two binding sites and doubled binding area.
  2. The interpretation presented in the paper under consideration stops at the moment which shall be the initiation for next step – probably the most important for immune responce – C1q complexation.
    >>Re: I agree that the events following Ab-Ag interaction and immune complex formation, such as C1q complexation, are very important in the humoral immune response. However, in this paper I intended to focus only on the binding energies and their distributions on a system level. In a previous paper, which is cited in this manuscript, the effector function of antibodies, mediated by cells and complement components was explained in the context of network formation and immunochemical reaction rate and physical aspects of binding.
  3. Is the model of universal character ? Is it applicable to any protein-protein complexation ?
    >> Re: I think the model is of universal character. For this reason, I tried to use general physical and biochemical terms besides the immunological terms. Whenever possible I described phenomena in general terms, in immunological terms, in network science terms, so as to show the applicability of the model in all areas.

Minor comments:

  1. The Figures are not sufficiently explained in legends. All symbols present on the figure shall be explained in legend – see Fig. 2 – only two maps on left are explained in legend to this figure. This comment is valid for all figures present in the paper.

Fig 3 – G – no explanation

General rule is to give as much information as it is necessary for the Reader to recreate the presentation. It is impossible in the case of the presented paper.
>> Re: The revised version contains more detailed legends for all figures.

  1. Fig. 3. Self-organising „system” (Ab system) ……….- Is it just one Ab molecule and one Ag molecule – or we are taking abut the large set of Abs and Ags

„response to thermodynamic activity of Ag, so as to maintain the system” – to maintain what system ? – one protein interacting with the second one or the whole population ?

Last comment on the „system” usage „μ is the chemical potential of antigen, the distance from system average chemical potential μ0” System – just in one molecule/complex  or in the whole population of Abs and Ags.
>> Re: In all cases I am referring to a system of many interacting molecules. The average chemical potential is the average of all interacting molecules in the system, but each molecule has its own chemical potential, which is the distance from this average. In the revised version of the manuscript this is hopefully clearly stated.

  1. Fig 8 – the line with the G – not explained in the legend
    >> Re: Legends of all figures were revised and extended to explain all letters. Figure 8 was slightly modified and G is not present as it was not really necessary.

Reviewer 2 Report

This paper describes a theoretical model for the adaptive immune system. Specifically, the model is based on chemical thermodynamics and network theory, and defines the adaptive immune system as a configuration space. Then the manuscript uses the description of the system to derive hierarchical networks governing antigen transport. Overall, the article shows that adaptive immune system can be modeled as complex physical system and that it obeys the laws chemical thermodynamic system. The manuscript also identifies attributes of the humoral immune system that capture the system dynamics. The paper is generally well written though the paper would benefit from simplifying language to make it more accessible to the general reader. The ideas presented are quite interesting and would be of interest to researchers in the field. I don’t have any major comments. My only suggestion is to revise the language with accessibility for the general reader in mind.

Author Response

I wish to thank the reviewer for considering my manuscript, my response is below the review.

This paper describes a theoretical model for the adaptive immune system. Specifically, the model is based on chemical thermodynamics and network theory, and defines the adaptive immune system as a configuration space. Then the manuscript uses the description of the system to derive hierarchical networks governing antigen transport. Overall, the article shows that adaptive immune system can be modeled as complex physical system and that it obeys the laws chemical thermodynamic system. The manuscript also identifies attributes of the humoral immune system that capture the system dynamics. The paper is generally well written though the paper would benefit from simplifying language to make it more accessible to the general reader. The ideas presented are quite interesting and would be of interest to researchers in the field. I don’t have any major comments. My only suggestion is to revise the language with accessibility for the general reader in mind.

>> Re: I thank the reviewer for reading and evaluating the manuscript. In the revised version I tried to simplify the abstract and some sections with the aim of rendering the text more intelligible for the general reader. All figure legends have been extended to provide a detailed description of what is shown. Additionally, I plan to write explanatory articles referring to this paper, so that the general readers will find summaries and plain language explanations for the presented ideas.

Reviewer 3 Report

Abstract is too much long try to summarize it

Add major contribution points at the end of introduction

Add literature study section after introduction

Add some other sections about the field of study 

All equations need to be explained in detail

Add results and discussion section 

Explain every figure in detail

Add some more details about future directions in conclusion

Author Response

I wish to thank the reviewer for reading and evaluating the manuscript. Considering the remarks of all three reviewers I revised the whole manuscript, with the general aim of better explaining all figures and equations, simplifying language and focusing on the key messages. Please find my poin-by-point responses below.

Abstract is too much long try to summarize it
>> Re: I shortened and simplified the abstract.

Add major contribution points at the end of introduction
>> Re: The key contribution was summarized at the end of introduction.

Add literature study section after introduction
>> Re: The most relevant papers in the literature are cited in the introduction and also in the Results and Discussion section. Since this is not an experimental paper, the theoretical considerations are continuously discussed and literature is cited throughout the paper.

Add some other sections about the field of study 
>> Re: The arrangement and the titles of the sections were slightly modified during revision to help the reader follow the logic of the approach. The present paper is close to a length limit that is readable, which I am afraid to make even longer, and it hopefully contains all sections and information required for understanding.

All equations need to be explained in detail
>> Re: more explanation is provided in the revised text for most equations.

Add results and discussion section 
>> Re: A combined “Results and Discussion” section now contain subsections with logical separation of the phenomena.

Explain every figure in detail
>> Re: The revised manuscript has extended legends and more detailed explanations for all figures.

Add some more details about future directions in conclusion
>> Re: The key future direction is the quantitative measurement of serum antibody interactions and the development of quantitative databases of antibody-antigen interactions. This is now included in the conclusions.

Reviewer 4 Report

The manuscript is of great interest for the mathematical description and understanding of how the immune system works. The article needs a little editing. The author should also carefully check all the designations in the field of drawings and captions to the drawings. For example, Figure 8 (B) is not described in any way in the caption under the figure. There are also edits to the text that the author did not have time to accept.

Comments from the point of view of scientific content rather bear the nature of wishes. So, it is known that in addition to affinity, there is avidity of antibodies and the description of the phenomenon of cooperative binding of two or more antibodies on an antigen molecule may be important. (see, for example, the work of Aristov et al. The use of statistical entropy in some new approaches to the description of biosystems. Entropy 2022, 24, 172). The article can be accepted after a little revision.

Author Response

The author thanks the reviewer for the time and efforts dedicated to this manuscript.

The caption of figure 8 was corrected to include reference to part B. Changes to the text were done using the “track changes” function to allow reviewers to follow the changes; the pdf version of the manuscript reflects the final form, with all changes accepted.

The author agrees that the avidity of antibodies is an important and interesting issue. The current model focuses on the steady state of the immune system, where a balancing between B-cell activation, Ab-Ag interaction and AbAg complex removal maintains quasi-static levels of these components. In this scheme antibody avidity probably plays an important role in the effector functions, therefore in the pathway that leads to the establishment of the steady state, but it plays a less obvious role in the steady state itself - at least according to the current presented model. The avidity of a particular antibody isotype can be an important determinant of the chemical potential of the antibody in steady state, but that factor does not appear in the distribution of chemical potentials, reflecting affinity of Fab fragments, itself.

This interpretation, of course, remains to be supported by experimental means and in the future further improvements to the model can be made to include cooperative models of binding if required, as suggested by the reviewer.

Round 2

Reviewer 1 Report

I see a big efford to correct the paper.

However I do not see the definition of "system"

I do not see the defintion of "self-assembly"

The main point

What is the difference between the Ab-Ag complexation in respect to  any other protein-protein complexation. I do not see the explanation whether the model is universal for the phenomenon of any form of protein-protein complexation.

The legends for figures are still missing the explanation of the symbols used in Figures.

Author Response

I thank the Reviewer for all the efforts of improving this paper. My responses are below the Reviewer's remarks.

However I do not see the definition of "system"

>> Re: In the manuscript there are several sections and sentences that define the system.

In "2.1 Compartments of the system" the overall approach to creating an abstract space from the interactions of the humoral immune system is described: lines 78-80 "Throughout the article we shall examine the adaptive immune system in an abstract space called configuration space. Configuration here refers to the arrangement of all the constituent Ab in a system of interacting molecules." Antibodies are the molecular entities that create the abstract space because the free energy changes occurring via interactions position them in the configuration space.

Line 116 "The system is composed of Ab and Ag molecules. " We are considering the humoral immune system, focusing on the interactions of antibodies and antigens, arranging all the interactions in the abstract configuration space, according to the free energy change accompanying the reaction and according to the specificity of the reaction.

Overall, the system in a broad sense is humoral immunity, in a strict sense it is the configuration space of antibody-antigen interactions.

I do not see the definition of "self-assembly"

>> Re: What is proposed in the paper is better described by self-organization. "Self-assembly" is an important concept but is primarily used in material science, referring to the process of organization of components without external direction. "Driven self-assembly" is associated with the flow of matter and energy, but is also primarily used for nanoparticles and nanomaterials. The reason I prefer self-organization is because the system, the immune system, actively shapes (genetic changes, structural changes, molecular fraction changes) the organization of the system via tuning its components. Local interactions are interconnected via the network architecture of the system.

The process of self-organization is described under "2.2 Principle of self-organization". The process is an immunological mechanism: cells are involved, because it is the cells that can bring about the above listed changes. Cells can sense the chemical potential of antigen, because they are equipped with membrane antibodies that are able to bind antigen and trigger proliferation and differentiation. Class-switching and somatic hypermutation are genetic changes that occur in cells, but modify the structure of antibodies. Cells secrete these antibodies, thereby adjusting their concentrations. Finally, it is also cells that remove antibodies along with bound antigen.

So in this aspect, cells are also part of the system, even though we only consider antibodies, antigens and their interactions in the strict definition of system.

What is the difference between the Ab-Ag complexation in respect to any other protein-protein complexation. I do not see the explanation whether the model is universal for the phenomenon of any form of protein-protein complexation.

>> Re: The word "complexation" is not used in the manuscript, because it is mostly associated with chemical processes. Antibody-antigen complexes are formed when these molecules interact and form non-covalent bonds. But this manuscript assumes that these complexes are swiftly removed from the system via Fc receptors. Of course, we are only looking at a certain step of the very complex series of events of a humoral immune response, but this is necessary for a model. It is not possible to incorporate all the immunological events into a mathematical model.

The key difference between antibody-antigen interactions and protein-protein interaction is that the immune system adapts. This is the unique property of vertebrate immune systems: accelerated evolution that constantly shapes the repertoire of antibodies, adjusting the interactions so as to adjust antigen concentrations. In this aspect, this model cannot be universal for protein-protein interactions.

On the other hand, I think the model is universal for adaptive processes. Biological adaptation is strongly dependent on proteins, on the immense conformational diversity the protein world possesses. It is the basis of cellular life, of cellular adaptations, which in turn provide the basis of higher-level adaptations to the surroundings, to the environment. Therefore, the model is potentially applicable for protein-protein interaction on an evolutionary time-scale.

The legends for figures are still missing the explanation of the symbols used in Figures.

>> Re: Further annotations were added to Figure 5, Figure 6, Figure 7 and Figure 8.

Round 3

Reviewer 1 Report

I have not received answers to my questions as to the "system" definition "self-assembly" definition and many others.

Conslusions are very poor and not convincing

complex biological system follow the rules that apply to a thermodynamic system

What is "system" - the sentence from Conclusions " complex biological system follow the rules that apply to a thermodynamic system"

However the problem discussed is of hight importance.

Author Response

The author thanks the reviewer for the time and efforts dedicated to this manuscript.

The expression "self-assembly" is not used in the article and the author feels that introducing it in the article would not in any way help its understandability. If it is not used, its definition can probably also be avoided without impairing the value of the paper; avoiding technical terms that are not indispensable may keep the text as simple as possible.

The thermodynamic system within the complex biological system of immunity can be defined as an open, multicomponent system of molecular interactions between antibodies and antigens. As compared to a physical system with very clear boundaries, e.g. a thermally insulated bottle with gas molecules filling it, the biological system can be thought of as treating only a part of the biological system as the thermodynamic system. Thus, it has no strict physical boundaries: antibodies and antigens interact everywhere in the organism, with some privileged sites, as the blood plasma, the lymphoid organs, the mucosal surfaces. The boundary is present only in the abstract theoretical space, because we consider only these interactions. Because of this abstraction, and because this system is embedded in a highly complex and dynamic environment, the model only serves to obtain the general laws and global variables of the system. This is why we regard the system an open one. Unlike the case of the thermos bottle or chemical reactions taking place in a test tube, no precise calculations can be made for all the interacting components. Yet, the validity of the model can be supported by the analogies described in the paper and by the experimental technologies that examine a single antigen reactivity. Of the multicomponent system, a selected subset of components can be measured by serological assays.

The conclusions were rewritten and supplemented with the above definition.
